# Machilin A Inhibits Tumor Growth and Macrophage M2 Polarization Through the Reduction of Lactic Acid

**DOI:** 10.3390/cancers11070963

**Published:** 2019-07-09

**Authors:** Tae-Wook Chung, Eun-Yeong Kim, Chang Woo Han, So Young Park, Mi Suk Jeong, Dahye Yoon, Hee-Jung Choi, Ling Jin, Mi-Ju Park, Yun Ju Kwon, Hanna Lee, Keuk-Jun Kim, Kang Hyun Park, Suhkmann Kim, Se Bok Jang, Ki-Tae Ha

**Affiliations:** 1Department of Korean Medical Science, Healthy Aging Korean Medical Research Center, Pusan National University, Yangsan, Gyeongsangnam-do 50612, Korea; 2Graduate Training Program of Korean Medicine for Healthy-aging, Pusan National University, Yangsan, Gyeongsangnam-do 50612, Korea; 3Department of Molecular Biology, College of Natural Sciences, Pusan National University, Busan 46241, Korea; 4Department of Chemistry, Center for Proteome Biophysics and Chemistry Institute for Functional Materials, Pusan National University, Busan 46241, Korea; 5National Development Institute of Korean Medicine, Gyeongsan, Gyeongsangbuk-do 38540, Korea; 6Department of Clinical Pathology, DaeKyeung University, Gyeongsan, Gyeongsangbuk-do 38547, Korea

**Keywords:** machilin A (MA), Warburg effect, cancer metabolism, lactate dehydrogenase, hypoxia

## Abstract

Lactate dehydrogenase A (LDHA) is an important enzyme responsible for cancer growth and energy metabolism in various cancers via the aerobic glycolytic pathway. Here, we report that machilin A (MA), which acts as a competitive inhibitor by blocking the nicotinamide adenine dinucleotide (NAD) binding site of LDHA, suppresses growth of cancer cells and lactate production in various cancer cell types, including colon, breast, lung, and liver cancers. Furthermore, MA markedly decreased LDHA activity, lactate production, and intracellular adenosine triphosphate (ATP) levels induced by hypoxia-induced LDHA expression in cancer cells, and significantly inhibited colony formation, leading to reduced cancer cell survival. In mouse models inoculated with murine Lewis lung carcinoma, MA significantly suppressed tumor growth as observed by a reduction of tumor volume and weight; resulting from the inhibition of LDHA activity. Subsequently, the suppression of tumor-derived lactic acid in MA-treated cancer cells resulted in decrease of neovascularization through the regulation of alternatively activated macrophages (M2) polarization in macrophages. Taken together, we suggest that the reduction of lactate by MA in cancer cells directly results in a suppression of cancer cell growth. Furthermore, macrophage polarization and activation of endothelial cells for angiogenesis were indirectly regulated preventing lactate production in MA-treated cancer cells.

## 1. Introduction

Lactate dehydrogenase (LDH) is a tetrameric enzyme composed of four subunits. The two most common subunits are LDHA and LDHB, and five isoenzymes of LDH are known: LDH-5, A4; LDH-4, A3B1; LDH-3, A2B2; LDH-2, A1B3; LDH-1, B4 [1,2]. LDH-5 is the most efficient isoenzyme for catalyzing the transformation of pyruvate to lactate [1]. Furthermore, it is known that the levels of LDHA expression are higher in many cancers than in normal tissues. Thus, the enhanced production of LDH-5 with four LDHA subunits in cancer cells leads to an increase in the glycolytic metabolism in the presence or absence of oxygen [1,2]. Interestingly, most cancer cells take up more glucose than normal cells, and depend on lactate fermentation by LDHA, which catalyzes conversion of pyruvate to lactate during the aerobic glycolysis known as ‘Warburg effect’ instead of oxidative phosphorylation for adenosine triphosphate (ATP) production and cell proliferation [3,4,5,6]. The 1,4-dihydronicotinamide adenine dinucleotide (NADH)-dependent enzyme LDHA mediates a redox reaction for the reduction of pyruvate to lactate at the end of glycolysis when the cytosolic NADH/nicotinamide adenine dinucleotide (NAD+) ratio is high, which is a crucial step for regenerating NAD+, and is needed to maintain glycolysis for the survival of cancer cells [6,7,8]. Metabolic changes of the Warburg effect are promoted by up-regulating the expression of the glycolytic gene *LDHA* during malignant transformation through oncogenic signals such as phosphoinositide 3-kinase (PI-3K) and 5’ adenosine monophosphate-activated protein kinase (AMPK) and oncogenic transcription factors such as Myc and Hif-1 [2,9,10]. Furthermore, lactate secreted by cancer cells into the tumor microenvironment is strongly associated with tumor-associated inflammation, immune evasion, and tumor angiogenesis for tumor progression [11]. In a previous study on genetic and pharmacologic targeting of LDHA, a knock-down of LDHA expression resulted in a decrease of mitochondrial membrane potential and suppressed carcinogenicity. Inhibition of LDHA and LDHA short hairpin ribonucleic acid (shRNA) induced oxidative stress, inhibited tumor progression, and suppressed tumor-initiating cell survival and proliferation [2,8,12,13,14]. Thus, LDHA may be a potential target for cancer therapeutics due to the suppression of the lactate metabolism. 

In the present study, we found out that machilin A (MA) as the strongest inhibitor on enzymatic activity of LDHA among 480 compounds in a natural product library. MA inhibited the LDHA activity through the suppression of an NADH cofactor binding to lactate dehydrogenase, based on structural and molecular investigations. This compound also suppressed the growth of tumor cells in in vitro and in vivo experiments by regulating LDHA activity. Furthermore, angiogenesis mediated by lactate produced and secreted from cancer was inhibited by MA. From these results, we suggest that MA, a novel LDHA inhibitor, may be candidates for anti-cancer drugs. 

## 2. Results

### 2.1. MA Inhibits In Vitro LDHA Activity by Binding of the Cofactor NADH to LDHA

The potential LDHA-inhibiting effects of 480 compounds obtained from National Development Institute of Korean Medicine (NIKOM) were examined, and MA produced the strongest inhibition of LDHA activity (Figure 1A). MA substantially inhibited LDHA activity in a dose-dependent manner (Figure 1B), and showed competitive inhibition of LDHA activity (data not shown). The inhibitory concentration (IC_50_) of MA for LDHA activity was 84 μM. MA consists of two heterocyclic compounds containing a methylenedioxy functional group (a 1,3-benzodioxole), therefore we tested whether commercially available 1,3-benzodioxole and piperonyl alcohol including a 1,3-benzodioxole structure has an inhibitory effect on LDHA activity. However, 1,3-benzodioxole and piperonyl alcohol did not affect LDHA activity up to a concentration of 1 mM (Appendix A).

LDHA activity is typically increased in various cancer cells, and tumor hypoxia catalyzes the conversion of pyruvate and NADH to lactate and NAD^+^. Furthermore, the tetrameric form of LDH exhibits stronger catalytically activity than the dimeric form [6]. Thus, to determine whether MA inhibits binding of the cofactor NADH to LDHA, MA and purified LDHA were incubated with Cibacron Blue 3GA-Agarose which resembles NADH and is a pseudo-affinity dye ligand of LDHA [6,15]. Both, NADH used as a positive control and MA prohibited LDHA binding to Cibacron Blue Agarose beads (Figure 1C). To further investigate whether MA would prevent the formation of tetrameric LDHA to inhibit LDHA activity, MA and purified LDHA were incubated in a crosslinking assay to confirm the occurrence of tetrameric LDHA. However, MA did not affect the suppression of LDHA tetramer formation (Figure 1D). 

### 2.2. LDHA and Its NADH Binding Structures are Similar and In Closed Active-Site Conformation

To further investigate binding ability of LDHA to MA/NADH, surface plasmon resonance (SPR) biosensor and isothermal titration calorimetry (ITC) analysis were performed (Figure 2A and Appendix A). Structural properties and interactions of full-length LDHA and MA were studied, and structural characterizations of LDHA with NADH and malonate were produced. Full-length LDHA is composed of 331 amino acids and includes a nucleotide binding region (residues 28–56; Appendix A). We determined the human LDHA structure with malonate at a resolution of 2.9 Å, and malonate was derived from crystallization conditions (Figure 2D and Appendix A). Statistics are shown in Appendix A. LDHA with malonate is a typical Rossmann-fold protein of mixed α/β secondary structures with a protruding N-terminal portion (Appendix A). This structure was solved in a new space group of monoclinic *P*2_1_, and four malonate molecules were bound to each LDHA tetramer. Recently, an apo-LDHA structure was solved in space group *P*4_1_22 or *P*3_1_21 [16,17]. According to our results, each monomer LDHA with malonate has one active catalytic substrate binding site (Appendix A). Two malonates are located in the active-site loop (residues 97–108) of the LDHA tetramer. LDHA with malonate in the substrate pocket has an active closed conformation (Appendix A). Oxygen atoms of malonate bind to the two charged residues (R105 and R168) and three polar amino acids (N137, H192, and T247) of LDHA.

The LDHA structure with NADH and malonate was solved at a 2.4 Å resolution, and the LDHA tetramer occurred in space group *P*2_1_ (Appendix A). Eight NADH molecules were in the large active co-substrate-binding pockets, and eight malonate molecules were located in small active substrate-binding pockets of the two LDHA tetramer. Each NADH binds to a monomer of the LDHA tetramer in about α6 and α10 helices (Figure 2D and Appendix A). Negatively charged NADH and malonate substrates were in the tunnel-type groove inside the positively charged surface (Figure 2C). The NADH adenosine and pyrophosphate unit were located at the opening of the tunnel cavity. The NADH nicotinamide moiety and malonate were within closed cavities, and they interacted with each other. In addition, NADH interacted with hydrophobic (A29, V30, A97, and V135), polar (N112, N137, S160, and H192), and charged residues (D51, K56, and R98) of LDHA (Figure 2C). Interaction distances of LDHA with NADH and malonate are shown separately (Appendix A). The active-site loop near NADH and malonate was in an active closed conformation (Appendix A). The structures of LDHA and its NADH binding site are similar and their root-mean-square deviation (RMSD) is 0.46 Å for main chain Cα atoms. Several minor structural deviations occurred in proximity of the N-terminal between the E15 and Q16 residues of LDHA.

### 2.3. MA Binds to LDHA In an Open Active-Site Conformation

The LDHA structure with MA (bis(3,4-methylenedioxy)-8,8’-lignan) was solved at 2.6 Å resolution in the space group *P*2_1_. The asymmetric unit comprised three LDHA tetramers containing one MA (Appendix A). Each LDHA tetramer contains two allosteric sites and four active sites [18]. In this study, one MA is located in an interfacial allosteric binding site between two monomers. To identify the conformation of the LDHA tetramer, we performed electron microscopy (EM) on the MA binding of LDHA (Appendix A). Despite the inclusion of MA the LDHA complex maintained the original tetramer-like conformation. MA binds to LDHA in an open active-site conformation (Figure 2D). The surface shape and charge of this LDHA including MA were shown. The oxygen of MA binds to S183 residue of LDHA through hydrogen bond (Figure 2C). The structures of LDHA with malonate and NADH were shown in an active closed conformation, whereas LDHA with inhibitor-MA showed in an inactive open conformation (Figure 2E). The root mean square deviation (RMSD) of the main chain Cα atoms between LDHA-MA and LDHA-malonate was 1.29 Å. The average RMSD between LDHA-MA and LDHA-NADH was 1.31 Å (Appendix A). Major structural deviations occurred around the active-site loop region of A97–N114 residues. Compared with the LDHA structure with malonate or NADH, the LDHA structure with MA induced large conformational changes in the active-site loop region, which is often referred to as the substrate-specificity loop.

### 2.4. MA Reduces Growth of Cancer Cells, Colony Formation, LDH Activity, Lactate Production and ATP level in Hypoxic Condition

To further investigate whether MA can suppress tumor growth, we preferentially tested cytotoxicity of MA. MA substantially inhibited the growth of HT29 and Hep3B cells under hypoxic and normoxic conditions (Figure 3A). Then, we performed colony formation assay with HT29 colon cancer cells and Hep3B liver cancer cells in the presence or absence of MA. MA significantly reduced the number of surviving colonies of HT29 and Hep3B cells (Figure 3B). We tested whether MA had an effect on the expression of LDHA under normoxic and hypoxic conditions. Although the expression of LDHA was not affected by MA treatment under normoxic and hypoxic conditions (Figure 3D), MA significantly reduced the increase of LDHA activity, lactate production, and adenosine triphosphate (ATP) synthesis through induced LDHA expression in hypoxic condition and the inhibition of LDHA activity and lactate production under normoxic condition (Figure 3C). In order to assess whether MA can inhibit LDH activity in various cancer cells, we examined the production of lactate and intracellular enzyme activity of LDH in MA-treated cancer cells. MA produced a considerable decrease in LDH activity in colon, breast, lung, and liver cancer cells, and lactate production was inhibited by MA (Table 1 and Table 2) in normoxic conditions. Moreover, MA significantly suppressed the growth of several cancer cells (Table 1), based on the inhibition of LDHA activity.

Previous studies reported that the induction of apoptosis by LDHA inhibition was associated with the production of mitochondrial reactive oxygen species (ROS) [2]. As shown in Figure, MA induced mitochondrial ROS and apoptosis (Figure 3E,F). In addition, by knockdown of LDHA expression by shRNA, the inhibitory effects of MA on LDHA activity, lactate production, and mitochondrial ROS level were abolished (Appendix A). These results suggest that the inhibitory effect of MA on the Warburg effect is mainly due to inhibition of LDHA enzymatic activity.

### 2.5. MA Suppresses LDHA Activity and In Vivo Growth of Lung Cancer LLC and CT26 Cells

First, we examined the inhibitory effect of MA on LDHA activity using murine Lewis lung carcinoma LLC (Appendix A). To investigate a potential inhibitory effect of MA on tumor growth in vivo, mice were subcutaneously injected with LLC and CT26 cells. MA was abdominally injected at a concentration of 1, 10, or 50 µg/g body weight. MA significantly reduced volume, weight, and size of LLC and CT26 tumors (Figure 4A,D, respectively). Furthermore, we investigated whether MA could affect activity and expression of LDHA in the tumor tissues of mice inoculated with LLC cells. The MA treatment clearly inhibited the activity of LDH in LLC cell allografts however it did not affect the expression of LDHA (Figure 4B,C). In addition, we also assessed in vivo toxicity of MA to liver and kidney, and no significant toxic effect of MA on liver or kidney in mice with allograft tumors was observed (Appendix A). These results suggest that MA can inhibit in vivo tumor growth by decreasing LDH activity without adverse effects on liver or kidney functioning. 

### 2.6. MA Inhibits Capillary Like-Tube Formation of Endothelial Cells Induced by Tumor-Derived Lactate Stimulated Macrophages

Lactic acid secreted by cancer cells can induce the expression of the vascular endothelial growth factor (VEGF) in macrophages and the expression of Arg1 and CD206 (a major M2 phenotype marker of macrophages), which promotes tumor growth by inducing neovascularization [19,20]. Thus, to determine whether MA inhibits neovascularization and M2 phenotype of macrophages in tumor microenvironment, we examined expression of CD31, a vascular endothelial marker, and CD206, a main M2-type macrophage marker, in LLC tumors of mice. As shown in Figure 5A, MA suppressed the formation of new vessels in tumors of inoculated mice (Figure 5A). Furthermore, MA markedly inhibited the expression of CD206 in macrophages in tumor tissues of mice, despite there was no change in the presence of macrophages that infiltrated tumor tissues (Figure 5B). However, MA did not directly inhibit growth and tube formation of endothelial cells and growth of RAW264.7 macrophage cells (Appendix A). 

Interestingly, the expression of Arg1 in macrophages was increased substantially by lactate secreted by cancer cells under hypoxia (Appendix A), but only slightly under normoxia. The concentration of lactate in culture media of HT 29 colon cancer cells was about 5.5 mM under hypoxia (Appendix A). Furthermore, we examined that lactate concentrations of ≥5 mM can substantially affect the regulation of the M2-like phenotype, *Arg-1*, *CD206*, *VEGF*, and *YM1* genes, of macrophages (Appendix A). The treatment of macrophages with medium harvested from MA-treated cancer cells in hypoxic condition showed a reduction of Arg1, CD206, YM1, and VEGF expression compared to the medium harvested from cancer cells in hypoxic condition. In contrast, Arg1, CD206, YM1, and VEGF expression was reduced in macrophages treated with medium harvested from MA-treated cancer cells in hypoxic condition (Figure 5D and Appendix A).

We also examined whether the change of macrophage phenotype through the regulation of lactate secretion from MA-mediated cancer would affect capillary like-tube formation of endothelial cells and angiogenesis in the tumor microenvironment. Tube formation of endothelial cells increased significantly when endothelial cells were treated with conditioned medium from macrophages after treatment of macrophage with medium collected from cancer cells under hypoxic conditions, compared to treatment of endothelial cells with conditioned medium from macrophage after treatment of macrophage with medium harvested from cancer cells under normoxic conditions (Figure 5E). These results suggest that MA reduced tumor growth by suppressing angiogenesis through the regulation of lactate-mediated macrophage phenotype expression in the tumor microenvironment. 

## 3. Discussion

The activation of oncogenic proteins stimulates glycolytic metabolism in tumor tissue [2,10]. Furthermore, increasing Hif-1 levels under non-hypoxic conditions can elicit the Warburg effect in cancer cells [3]. Hypoxia also increases stability of the main factor Hif-1 for adaptation to hypoxic stress, a common feature in microenvironment of solid tumors, which induces the expression of glycolytic enzymes including LDHA (due to the Warburg effect), which again results in an increase of lactate production and cancer cell survival [2,10]. Thus, LDHA affects initiation, maintenance, and progression of cancers [8,14]. In the present study, we screened 480 chemical compounds for their potential inhibitory effects on LDHA activity. Interestingly, MA prohibited NADH from binding and inhibiting LDHA, and therefore inhibited growth of various cancer cell types. Furthermore, our findings demonstrated that the suppression of lactate production and secretion in cancers treated with MA under hypoxic conditions resulted in an inhibition of angiogenesis through the regulation of macrophage polarization in the tumor microenvironment. 

Many cancer cells use the aerobic glycolysis pathway instead of oxidative phosphorylation for ATP production, rapid macromolecule biosynthesis, and cell growth [14]. Furthermore, under hypoxic conditions, accelerated glycolysis ensures that the produced ATP levels meet the demands of fast proliferating tumor cells in low-oxygen environments [14]. In contrast, depletion of the glycolytic enzyme LDHA and of the LDHA inhibitor FX11 reduced ATP levels and induced significant oxidative stress and cell death [2]. Furthermore, reduction of LDHA resulted in a stimulation of mitochondrial respiration and a decrease of mitochondrial membrane potential, and interfered with carcinogenicity of malignant cells [14]. Previous studies reported that the mechanism involved in the induction of apoptosis by LDHA inhibition were as associated with the production of mitochondrial ROS [2,21]. Our results showed that the inhibition of LDHA activity by MA treatment reduced lactate and ATP production under normoxic and hypoxic conditions, leading to the suppression of tumor growth in vitro and in vivo (Figure 3 and Figure 4). Furthermore, MA induced apoptosis and production of mitochondrial ROS via an inhibition of LDHA activity (Figure 3E). However, MA did not affect LDHB activity when used in concentrations of up to 500 μM (data not shown).

Previously, MA have been reported that it has anticancer effect by inhibiting phospholipase Cγ1 [22] and by activating caspase-3 activity [23]. MA also has activity of inhibiting melanin synthesis [24], protecting liver damage [25], and activating osteoblast differentiation [26]. In addition, MA has selective inhibitory effect on drug metabolism via directly interacting with human liver cytochrome P450 1A1 (CYP1A1) and 2B6 [27]. These enzymes are involved with phenacetin *O*-deethylation and bupropion hydroxylation, respectively. Although these enzymes are not directly related with survival or malignancy of cancer, these enzymes are involves in the bioactivation and detoxification of carcinogen [28]. Thus, to identify a selective LDHA inhibitor, more extensive studies on MA and its derivatives is need. 

In our structural study, the X-ray crystal structure of purified human LDHA showed a tetrameric complex consisting of 4 LDHA subunits (LDH-5). However, MA did not disrupt the formation of active tetrameric LDHA to inhibit LDHA activity (Appendix A). In addition, MA did not prevent LDHA tetramer formation, as observed using a purified human LDHA crosslinking assay (Figure 1D). For LDHA activity, the NADH cofactor binds to LDHA. Subsequently, pyruvate interacts as a substrate with a binary LDHA/NADH complex, which catalyzes the conversion of pyruvate to lactate [29]. In this study, LDHA interacted with NADH biomimetic beads. However, MA inhibited LDHA binding to the NADH mimic (Figure 1C). Additionally, surface plasmon resonance (SPR) biosensor and isothermal titration calorimetry (ITC) analysis were performed to measure the binding ability of LDHA to MA/NADH (Figure 2A and Appendix A). We found that MA physically bound to LDHA with an apparent K_D_ of 29 μM and NADH bound to LDHA with an apparent K_D_ = 10 μM.

To examine the conformation of LDHA tetramers, we performed electron microscopy on MA binding to LDHA (Appendix A). Images of negatively stained samples were assessed, and the complexes were identified as tetramer-like shapes. In the present study, we tested interactions and the molecular basis of full-length LDHA and its natural inhibitor MA. In addition, molecular characterizations of LDHA with NADH and malonate were compared. These results showed that MA binds to LDHA in an open inactive conformation (Figure 2D). Binding of MA to LDHA can decrease LDHA activity in lactate production by suppressing the binding between LDHA and the cofactor NADH. Interestingly, the oxygen of MA binds to S183 residue of LDHA through hydrogen bond (Figure 2C). In the more active form of the enzyme, phosphorylase *a*, specific Ser residues, one on each subunit, are phosphorylated [30]. LDHA with MA may make inactive conformation by blocking phosphorylation by binding of MA with Ser183. In complexes of LDHA and NADH or malonate, the polar LDHA amino acids N137 and H192 are important residues for interactions with substrates through hydrogen bonding. Interestingly, the structures of LDHA with malonate and NADH were shown with all active site loops in closed conformation, whereas when LDHA was associated with the inhibitor MA, the active site loops were in an open conformation. In a previous study on molecular dynamics of LDHA inhibitors, Shi et al. suggested that closure of the mobile loop is not necessarily required for ligand binding, and certain S-site binders may force the loop open when binding [31]. Compared to the LDHA structure with malonate or NADH, the structure of LDHA associated with MA showed substantial conformational changes in the active-site loop. The residues Q100–N107 in the loop region of the LDHA active-site with MA migrated to the open form over more than 12 Å from the closed form. These results suggest that considerable conformational changes of LDHA occurred in the active-site loop regions exposed to solvents in an open MA-binding form. These results provide insights into LDHA enzyme mechanisms and inhibition and provide framework for structure-assisted drug design that may contribute to new cancer therapies.

Lactate produced by cancer cells due to the Warburg effect results in acidification of the cancer’s microenvironment; lactate is also transported to neighboring cancer cells, immune cells, and vascular endothelial cells which promotes cancer development, tumor inflammation, angiogenesis, and metastasis [11]. In previous studies, elevated lactate levels in cancer cells due to enhanced expression of LDHA during hypoxia substantially increased the N-myc downstream regulated gene (NDRG)-3 protein expression, which induced the Raf-extracellular signal regulated kinase (Raf-ERK) pathway and thus promoted angiogenesis and hypoxic cell growth [32]. Furthermore, lactate metabolized by tumors modulates the phenotype expression of endothelial cells and thereby affects tumor vascular development [33]. It was previously reported that lactate produced by LDHA in cancer cells is a potent inhibitor of the function and survival of T and Natural killer (NK) cells, leading to tumor immune evasion [34]. Colegio et al. demonstrated that tumor-derived lactate affected M2 phenotype polarization of tumor-associated macrophages, which play an important role in tumor growth [19]. Moreover, it was shown that tumor-conditioned media or lactic acid up-regulated the expression of VEGF angiogenic factor and Arg-1 M2 phenotype marker in macrophages [19]. In line with these results, we found that tumor-derived lactate up-regulated the expressions of *Arg-1*, *CD206*, *VEGF*, and *YM1* genes in macrophages, and an MA-mediated decrease of tumor-derived lactate resulted in a reduction of *Arg-1*, *CD206*, *VEGF*, and *YM1* mRNA levels in macrophages (Figure 5D). In contrast, the reduced expression of *Arg-1*, *CD206*, *VEGF*, and *YM1* mRNAs mediated by MA was reduced by the addition of 5 mM lactate (Figure 5D). These results clearly suggest that MA inhibited M2-like phenotype polarization of macrophages through the suppression of lactate production by regulating cancer cell metabolism under hypoxic conditions. 

In the tumor microenvironment, immune cells are recruited into the tumor site for tumor progression [35]. Among them, macrophages are particularly abundant and are strongly associated with tumor initiation, progression, and malignancy although macrophages are also important for anti-tumor immunity [20,35], indicating the importance of macrophage M2 polarization by various factors in the tumor microenvironment. Tumor-associated macrophages (TAMs) accumulated in hypoxic regions of tumor [20] generally show various characteristics of the alternatively activated M2-like phenotype, and also play an important role in vascular programming of tumors by enhancing the production of the VEGF for promotion of angiogenesis [36,37]. In this study, the increase of angiogenic factors including VEGF and Arg-1 in macrophages due to tumor-derived lactate under hypoxic conditions affected the activation of endothelial cells for neovascularization and M2-like polarization of macrophages in vitro and in vivo (Figure 5), whereas the reduction of tumor-derived lactate by MA treatment under hypoxic conditions showed an inhibition of neovascularization and M2-like polarization of macrophages by suppressing VEGF and Arg-1 expression (Figure 5). 

In the previous study, we demonstrated that 1-(phenylseleno)-4-(trifluoromethyl)benzene (PSTMB) inhibited LDHA activity through reducing its binding to substrate, pyruvate, without affecting the expression of LDHA. In addition, PSTMB induced mitochondria-mediated apoptosis of cancer cells [38]. Consistent to the study, in this study we show that another LDHA inhibitor, MA, successfully suppressed their own growth of cancer cells. In addition, we further demonstrated that as a result of regulation of the Warburg effect, MA also inhibited the production of lactic acid released from cancer cells and suppressed the macrophage-mediated neovascularization. 

Several small molecule LDHA inhibitors, such as FX11, galloflavin, gossypol, and GNE140, are under estimating for developing novel and effective anticancer agents [39,40,41]. However, due to the limitation of their intracellular uptake, pharmacokinetic profile, and in vivo activity, still no LDHA inhibitor used for anticancer agents in clinic [39]. Because MA has a unique structural feature compared with previously reported LDHA inhibitors, we suggest that MA or its derivatives might be good candidates for developing a novel anticancer drug through improvement in selectivity, activity, delivery, and pharmacokinetic profiles. In addition, several reports demonstrated that Warburg effect is closely related with aggressiveness and drug resistance of cancer cells [42,43]. Furthermore, resistance to LDHA inhibitor was also overcome by addition of phenformin, an inhibitor of mitochondrial complex I [41]. From these results, MA might be developed as a combinative therapeutic agent to resistant cancers. 

## 4. Materials and Methods

### 4.1. Materials

A chemical library of 480 compounds was obtained from National Development Institute of Korean Medicine (NIKOM). Antibodies for glyceraldehyde 3-phosphate dehydrogenase (GAPDH) were purchased from Santa Cruz Biotechnology (sc-32233, Dallas, TX, USA). The anti-LDHA antibody was obtained from Santa Cruz Biotechnology (sc-137243). All chemicals and reagents, including NADH, pyruvate, Cibacron Blue agarose, 1,3-benzodioxole, piperonyl alcohol, and nordihydroguaiaretic acid were obtained from Sigma-Aldrich (St. Louis, MO, USA). 

### 4.2. Assay Compounds 

Compounds such as MA, 1,3-benzodioxole, piperonyl alcohol, and nordihydroguaiaretic acid are typically dissolved in dimethyl sulfoxide (DMSO). Compounds with limited aqueous solubility were dissolved in DMSO before adding to the respective medium or buffer, and the final concentration of DMSO in cell culture media and assays was 0.1% (v/v).

### 4.3. LDH Activity Assay

A purification of recombinant LDHA was performed as described previously [6,44]. LDH activity was determined by measuring the decrease of absorbance at 340 nm due to the oxidation of NADH in 20 mM HEPES-K^+^, 0.05% bovine serum albumin (BSA), 20 μM NADH, and 2 mM pyruvate (Sigma-Aldrich, St. Louis, MO, USA) at a pH of 7.2 using a spectrofluorometer (Spectramax M2, Molecular Devices, San Jose, CA, USA; excitation 340 nm; emission 460 nm), as described previously [6]. Ten nanograms of recombinant LDHA protein were used in the in vitro LDH activity assay. One microgram of total protein from cell or tissue lysates was used for intracellular or in vivo LDH assays. 

### 4.4. NADH Binding Assay 

An NADH binding assay was performed as described previously [6]. The NADH binding ability of LDHA was determined by measuring the affinity of LDHA to agarose-immobilized Cibacron Blue 3GA, which resembles NADH. Then, 400 ng of purified recombinant human LDHA (rhLDHA) was incubated with NADH or MA, followed by incubation with 30 µL of Cibacron Blue agarose at 4 °C for 2 h. After a washing step with 20 mM Tris-HCl (pH 8.6), LDHA bound to beads was eluted in PBS using a sodium dodecyl sulfate (SDS) gel running buffer and was then subjected to–SDS-polyacrylamide gel electrophoresis (SDS-PAGE) and western blotting. Standardized amounts of protein were used in each reaction. 

### 4.5. Crosslinking Assay for Tetramer Formation of LDHA

The purified rhLDHA (100 ng) was incubated with or without MA at room temperature for 5 min in the presence of 0.0025% glutaraldehyde containing 5% glycerol, 10 mM Tris–HCl (pH 7.5), 50 mM NaCl, 0.1 mM ethylenediaminetetraacetic acid (EDTA), 1 mM dithiothreitol (DTT). After this, SDS sample buffer was added and the samples were subjected to SDS-PAGE. The same amounts of non-crosslinking proteins were used in each reaction. 

### 4.6. Surface Plasmon Resonance (SPR) Biosensor Analysis

The apparent dissociation constants (K_D_) between LDHA and Machilin A were measured using a Biacore T100 biosensor (GE Healthcare, Chicago, IL, USA). The purified LDHA was covalently bound to the Series S sensor chip CM5 using an amine coupling method as suggested by the manufacturer. A total of 150 μL LDHA (50 μg/mL) in 10 mM sodium acetate (pH 5.5) was coupled via injection for 15 min at 10 μL/min, followed by injection of 1 M ethanolamine to deactivate the residual amines. To measure kinetics at 25 °C, chemical compounds with concentrations ranging from 500 to 62 μM were prepared by dilution in running buffer (10 mM HEPES, 150 mM NaCl, 0.005% v/v surfactant P20, and 1% DMSO) at a pH of 7.4. The immobilized ligand was regenerated by injecting 10 μL of 50 mM NaOH at a rate of 10 μL/min during the cycles.

### 4.7. Expression and Purification of LDHA for Structural Study

Full-length LDHA (residues 1–331) was obtained from cDNA by polymerase chain reaction (PCR) and cloned between the *NdeI* and *XhoI* sites of the pET-28a (Novagen, Burlington, MA, USA) vector. The successfully cloned plasmid, as confirmed by DNA sequencing (performed by SolGent, Daejeon, Korea), was used to transform *E. coli* BL21 (DE3) cells. A single colony was cultured overnight at 37 °C. This culture was inoculated placed in Luria-Bertani medium containing kanamycin (100 mg/mL), and the cells were grown at 37 °C. Expression of LDHA was induced with 0.5 mM isopropyl β-D-1-thiogalactopyranoside (IPTG) for 16 h at 25 °C. The cells were centrifuged for harvesting, and the cell pellets were resuspended in lysis buffer A (50 mM Tris-HCl at pH 7.5 and 200 mM NaCl). The cells were subsequently disrupted by sonication on ice, after which the soluble LDHA supernatant was loaded onto a Ni-NTA column (Amersham Pharmacia Biotech, Buckinghamshire, UK). The column was washed with buffer A containing 20 mM imidazole, after which the bound protein was eluted using buffer A containing 200 mM imidazole. Purified fractions of the LDHA protein from the Ni-NTA column were mixed and purified using gel-filtration chromatography and fast protein liquid chromatography (FPLC) using a Superdex 200 10/300 GL column (Amersham Pharmacia Biotech, Buckinghamshire, UK) equilibrated in buffer A, after which the fractions were collected. Finally, the purified protein was concentrated to about 10 mg/mL. All purification steps were assessed using 15% SDS-PAGE and Coomassie Blue staining (Sigma-Aldrich, St. Louis, MO, USA). The purified LDHA was confirmed by a single band with a molecular weight of 39 kDa using SDS-PAGE.

### 4.8. Crystallization of LDHA, LDHA-NADH, and LDHA-MA Complexes

LDHA crystals were grown in hanging drops using a reservoir buffer of 200 mM sodium malonate and polyethylene glycol (PEG; 3.350). LDHA crystals with compounds were grown in hanging drops using a reservoir buffer (Tacsimate, pH 7.0, 20% PEG 3.350) with 50 mM compounds. The compounds used in this study were NADH and MA (Appendix A). Crystals appeared after three days and grew to maximum size within two weeks (Appendix A) at 20 °C.

### 4.9. Data Collection, Structure Determination, and Refinement

Data for full-length LDHA, LDHA-NADH, and LDHA-MA were collected at the Pohang Accelerator Laboratory (Pohang, Republic of Korea). Before data collection, each crystal was briefly soaked in a cryoprotectant solution consisting of the precipitant solution and 20% glycerol. The crystal was shock-cooled in a liquid-nitrogen stream at 100 K. Diffraction data were then produced and were processed using the *HKL*-2000 software package. The structures of LDHA, LDHA-NADH, and LDHA-MA complexes were determined using the molecular replacement method of the software programs *AMoRe, CCP4i* and *EPMR* [45,46]. The model was improved by iterative model building using the software *O* [47]. LDHA (amino acids 2–332; PDB entry 1I10) was used as an initial search model [48]. As shown in Appendix A, final refinements were performed using the *CNS* and *Phenix* software packages [49]. Structural visualizations shown in Figure 2 and Figure 3 were generated using MolScript, GRASP, Raster3D, and PyMOL [50,51,52,53]. Ramachandran statistics were calculated using Procheck. The atomic coordinates and structure factors (codes 5ZJE, 5ZJD, and 5ZJF) have been deposited in the Protein Data Bank.

### 4.10. Cell Culture, Hypoxia Condition and Determination of Cell Viability

Human colon cancer cells (WiDr, DLD-1, RKO, and HT29) and murine Lewis lung carcinoma (LLC) cells were purchased from the American Type Culture Collection (Manassas, VA, USA). Human breast cancer MCF7 cells, human hepatocellular carcinoma cells (Huh-7, HepG2, and Hep3B), and human lung cancer cells (HCC-95 and NCI-H1793) were provided by the Korean Cell Line Bank (Seoul, Korea). The HCC-95 and NCI-H1793 cells were grown in a Roswell park memorial institute medium (RPMI) 1640 medium (HyClone^TM^; GE Healthcare, Chicago, IL, USA). All other cells were grown in Dulbecco’s modified eagle medium (DMEM) supplemented with L-glutamine (200 mg/L), 10% (v/v) heat-inactivated fetal bovine serum (FBS; Sigma-Aldrich, St. Louis, MO, USA), and antibiotics (100 U/mL penicillin and 100 μg/mL streptomycin; Thermo Fisher Scientific, Waltham, MA, USA) in a humidified incubator at 37 °C and with 5% CO_2_ before the experiments. For induction of hypoxia condition (1% oxygen), the cells were cultured were cultured in the mixture of 94% nitrogen and 5% CO_2_/air at 37 °C for 24 h in cell culture incubator. Cytotoxicity of MA was evaluated using an MTT assay. Briefly, cells were incubated in 24-well plates with specified concentrations of MA for 24 or 48 h. Then, MTT solution (2.0 mg/mL) was added to each well. After 4 h of incubation at 37 °C and 5% CO_2_ in a cell culture incubator, the conditioned media were removed, and the amount of formazan crystals formed in living cells was estimated by measuring the absorbance at 540 nm using a microplate reader (Spectramax M2; Molecular Devices, CA, USA). The respective percentages of living cells were compared with a control treatment.

### 4.11. Colony Formation Assay

Base agar (0.6%) was placed at the bottom of the 6-well culture plate. After 30 min at room temperature, top agar (0.4%) mixed with HT29 (1 × 10^4^ cells) or Hep3B (1 × 10^4^ cells) cells was added to the base agar. HT29 and Hep3B cells were incubated at 37 °C and 5% CO_2_ for three weeks. The colonies were stained using 0.005% crystal violet for 1 h after which they were photographed and counted. 

### 4.12. Lactate Production Assay 

Cellular lactate production was measured using a lactate fluorometric assay kit (Biovision, Milpitas, CA, USA) as described previously [6]. Phenol red-free DMEM medium without FBS was added to a six-well plate containing subconfluent cells which were then incubated for 1 h at 37 °C. After incubation, the medium of each well was tested using a lactate assay kit (Biovision, Milpitas, CA, USA). 

### 4.13. Measurement of Intracellular ATP 

Intracellular ATP concentration was measured using an ATP bioluminescent somatic cell assay kit (Sigma-Aldrich) as previously described [6]. A total of 1 × 10^6^ cells were trypsinized and resuspended in ultrapure water. Luminescence was measured using a spectrofluometer (Spectramax M2, Molecular Devices, CA, USA) immediately after the addition of an ATP enzyme mix to the cell suspension. 

### 4.14. Western Blot Analyses

Total protein was extracted from each cell using 1% NP-40 lysis buffer (150 mM NaCl, 10 mM HEPES (pH 7.45), 1% NP-40, 5 mM Na pyrophosphate, 5 mM NaF, 2 mM Na_3_VO_4_) containing protease inhibitor cocktail tablets (Roche, Basel, Switzerland). Equal amounts (20 μg) of protein from each sample were fractionated using SDS-PAGE and the protein was then transferred onto nitrocellulose membranes (Hybond ECL; GE Healthcare, Chicago, IL, USA) using electrophoresis. The membranes were blocked for 1 h at room temperature using 5% non-fat dry milk and were then incubated with primary antibodies specific to the target protein at 4 °C overnight. The membranes were washed three times, and incubated with the respective secondary antibodies conjugated with horseradish peroxidase. The specific bands of targeted proteins were detected using ECL Plus (GE Healthcare, Chicago, IL, USA). 

### 4.15. Measurement of Mitochondrial Reactive Oxygen Species (ROS)

The production of mitochondrial ROS was determined by MitoSOX™ Red (Thermo Fisher Scientific, Waltham, MA, USA), respectively. Briefly, 5 μM MitoSOX™ Red was added to each cells cultured in conditioned medium and incubated at 37 °C for 10 min. The cells were then washed twice with PBS and the fluorescence intensity was analyzed using a BD FACS Canto II (BD Biosciences, San Jose, CA, USA) measuring excitation/emission at wavelength of 510/580 nm. 

### 4.16. Detection of Apoptotic Cells

HT29 cells were treated with the indicated concentrations of PSTMB for 48 h. Apoptotic cells were detected using the Annexin V-FITC Apoptosis Detection kit (Life Technologies, Carlsbad, CA, USA). The cells were suspended in 500 μL of buffer and treated with 5 μL of Annexin V-FITC. The fluorescence intensities were measured using BD FACSCANTO II (BD Biosciences, San Jose, CA, USA).

### 4.17. Tumor Allograft Model

LLC and CT26 cells were suspended separately in PBS (5 × 10^5^ cells/100 µL PBS) and inoculated in subcutaneously on the back of 7-week-old C57BL/6 mice and BALB/c mice respectively (Orient Bio Inc., Seongnam, Korea; n = 7). One day after the inoculation with tumor cells, mice of the control group (n = 7) were intraperitoneally administrated 100 µL of PBS per day for 18 days. Each of the MA-treated mice were intraperitoneally administrated 0.02, 0.2, or 1 mg MA, respectively, in 100 µL PBS per 20 g body weight, each day for 18 days. All mice were euthanized 19 days after inoculation with the tumor cells, and the tumors were excised and weighed. Length (l) and width (w) of each tumor were measured using a caliper, and the respective volume (V) was calculated according to the formula V = [(l × w^2^)/2]. Tumors were immediately removed from euthanized mice and prepared for histological examinations. To test liver and kidney toxicity of MA, at the end of the *in vivo* allograft experiment blood was collected from each mouse and serum was separated. Biochemical analyses of aspartate aminotransferase (AST), alanine aminotransferase (ALT), creatinine, and blood urea nitrogen (BUN) were performed using a commercial service (Green Cross Co., Yongin, Korea). All experimental procedures followed the Guidelines for the Care and Use of Laboratory Animals of the National Institutes of Health of Korea, and were approved by the Institutional Animal Care and Use Committee of Pusan National University, Republic of Korea (PNU-2018-1954, Date of approval: 27th June 2018).

### 4.18. Immunohistochemistry

Tumor specimens were immediately removed from euthanized mice, fixed in PBS with 3.7% formalin, and embedded in paraffin for immunohistochemical analyses. The paraffin sections were immunostained using an antibody specific to the endothelial cell marker CD31 (ab28364, Abcam, Cambridge, UK), macrophage marker CD68 (ab125212, Abcam, Cambridge, UK), and M2 phenotype macrophage marker CD206 (ab64693, Abcam, Cambridge, UK). The staining was visualized using a Dako EnVision kit (Dako, Santa Clara, CA, USA), and counterstained with hematoxylin.

### 4.19. Reverse-Transcription (RT)-PCR and Quantitative Real-Time RT-PCR (qRT-PCR)

Total RNA of the cells was isolated using a GeneJET RNA Purification Kit (Thermo Fisher Scientific, Waltham, MA, USA). One microgram of RNA was reverse-transcribed using a RevertAid reverse transcriptase (Thermo Fisher Scientific, Waltham, MA, USA), and single-stranded cDNA was PCR-amplified using AccPower® PCR PreMix (Bioneer, Daejeon, Korea). The qRT-PCR was performed using the QuantiNova™ SYBR®Green PCR Kit (Qiagen, GmbH, Hilden, Germany) and a Qiagen Rotor-Gene® Q Real-Time PCR Detection System according to the manufacturer’s instructions. PCR products were separated using electrophoresis on 1.5% agarose gel containing ethidium bromide in 1 × Tris-acetate buffer and were visualized under UV light. 

### 4.20. Tube Formation Assay

To investigate the formation of a capillary-like network of human umbilical vein endothelial cells (HUVECs) through the activation of tumor associated macrophages (TAMs) by lactate secreted by cancer cells, a tube formation assay was performed according to a previous study [54], with several modifications. For the preparation of conditioned media, HT29 colon cancer cells (3.5 × 10^5^ cells) suspended in 2 mL of DMEM were seeded on 6-well plates, with or without addition of 30 µM MA, and incubated under normoxic or hypoxic conditions. After 48 h of incubation, 1 mL of conditioned medium harvested from cancer cells was added to 6-well plates seeded with or without RAW264.7 macrophage cells (1 × 10^6^ cells), which were incubated at 37 °C and 5% CO_2_. After 24 h, 500 µL of conditioned medium were harvested from each well to examine tube formation. For preparation of HUVECs, Matrigel (13.9 mg/mL) was thawed at 4 °C and mixed with endothelial cell medium-basal (Endothelial Cell Medium (ECM)-b, ScienCell Research Laboratories, Carlsbad, CA, USA) at a ratio of 1:1. Seventy microliters of ECM-b-diluted Matrigel (6.95 mg/mL) was added to each well of the 24-well culture plates and allowed to polymerize at 37 °C for 1 h. HUVECs (ScienCell Research Laboratories) were detached from the tissue culture plates, washed, resuspended in the ECM-b medium containing 1% FBS, and seeded in the Matrigel-coated wells (5 × 104 cells/500 µL in each well).The harvested conditioned medium was added to the seeded HUVECs. After incubation for 12 h at 37 °C and 5% CO_2_, capillary-like tube formation of each well in the culture plates was photographed using a Nikon ECLIPSE TS100 light microscope (Nikon, Tokyo, Japan).

### 4.21. Statistical Analyses

Cell viability, LDHA activity, lactate production, and ATP production assays were calculated as a percentage of the respective value in control cells and expressed as mean ± standard deviation (SD). Differences between the mean values of the experimental groups were tested by a one-way analysis of variance (one-way ANOVA) and a student’s t-test using GraphPad Prism software (GraphPad Software, San Diego, CA, USA). Statistical significance is reported at *p* < 0.05. Each experiment was performed in three independent replicates.

## 5. Conclusions

We demonstrated that MA suppressed lactate dehydrogenase activity through the inhibition of NADH cofactor binding to lactate dehydrogenase. Thus, MA inhibited cancer growth and energy metabolism by inhibiting the activity of LDHA in cancer cells both under normoxic and hypoxic conditions. Consequently, the reduction of lactate released from cancer cells by MA influenced macrophage polarization and activation of endothelial cells for angiogenesis. From these results, we suggest that MA might be a potential candidates for the development of novel anticancer drug via inhibition of cancer metabolism

## Figures and Tables

**Figure 1 cancers-11-00963-f001:**
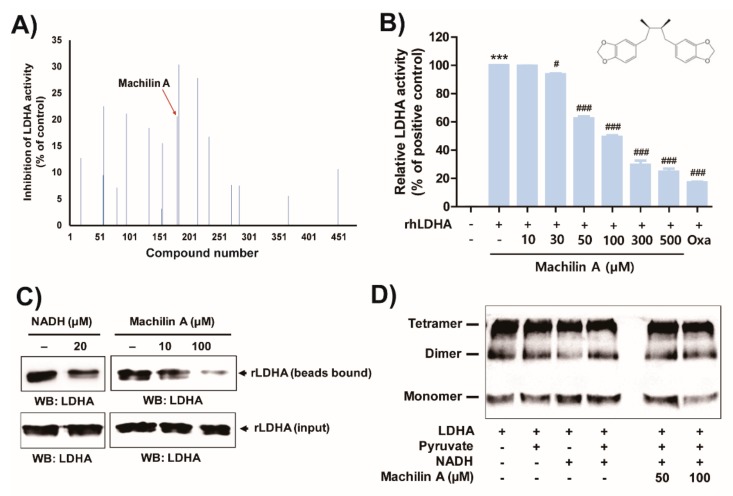
Inhibitory effect of machilin A on Lactate dehydrogenase A (LDHA) activity by binding of the 1,4-dihydronicotinamide adenine dinucleotide (NADH) to LDHA. (**A**) The screening on inhibition of LDH activity was analyzed using purified recombinant LDHA. (**B**) The inhibition of LDH activity was analyzed in presence of the indicated concentrations of machilin A. Oxamate (50 mM) was used as positive control for LDHA inhibition. Results of three independent replicates are presented as means ± SD. Statistic tests were performed using a one-way analysis of variance (ANOVA). *** *p* < 0.001 compared to the negative control (first column). # *p* < 0.05 and ### *p* < 0.001 compared to the positive control (second column). (**C**) The NADH binding ability of LDHA was determined by measuring the affinity of LDHA to agarose-immobilized Cibacron Blue 3GA, an NADH mimic. The purified recombinant human dehydrogenase A (rhLDHA; 400 ng) were incubated with Cibacron Blue agarose in the absence or presence of machilin A (10 and 100 μM). NADH was used for the positive control. LDHA that bound to beads was subjected to sodium dodecyl sulphate-polyacrylamide gel electrophoresis (SDS-PAGE) and western blotting. Equal amounts of protein were used in each reaction. (**D**) The purified recombinant LDHA (100 ng) with or without machilin A was incubated at room temperature for 5 min in the presence of 0.0025% glutaraldehyde containing 5% glycerol, 10 mM Tris–HCl (pH 7.5), 50 mM NaCl, 0.1 mM ethylenediaminetetraacetic acid (EDTA), and 1 mM dithiothreitol (DTT). The samples were subjected to SDS–PAGE.

**Figure 2 cancers-11-00963-f002:**
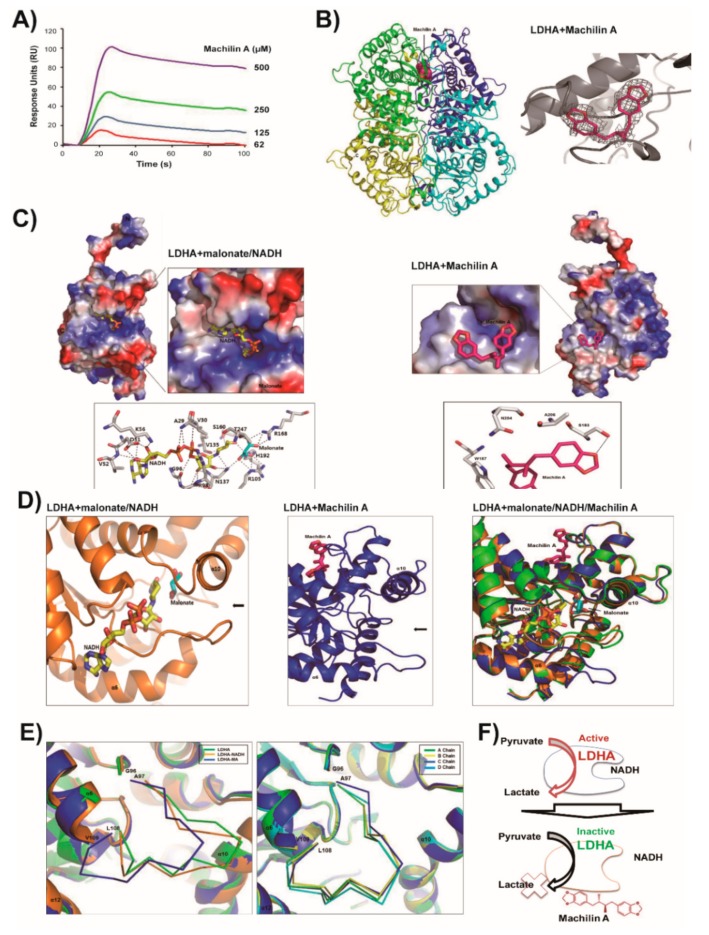
Surfaces and detailed interactions of Lactate dehydrogenase A (LDHA) and malonate/1,4-dihydronicotinamide adenine dinucleotide (NADH)/Machilin A. (**A**) Surface plasmon resonance (SPR) biosensor analysis of LDHA binding to machilin A at 25 °C. Machilin A sensorgrams for 62, 125, 250, and 500 µM are shown. (**B**) Tetrameric structure of human LDHA. Chains A, B, C, and D of LDHA are shown in green, yellow, blue, and cyan, respectively. The 2*F_o_-F_c_* electron density maps of LDHA with machilin A are contoured at 1σ (gray) (**C**) Shown are the complexes and side chains of several amino acids of LDHA and malonate/NADH (left panel) and LDHA and machilin A (right panel). The relative distribution of the surface charge is shown with the acidic region in red, the basic region in blue, and the neutral region in white. Hydrogen bonds in LDHA complexes with malonate/NADH, and machilin A, respectively, are shown as black dotted lines. (**D**) The active-site loop regions of LDHA-malonate/NADH and machilin A are shown as closed and opened forms for substrate binding, respectively (left and middle panels). Superpositions of the LDHA complexes with malonate, NADH, and machilin A are shown (right panel). (**E**) Closed active-sites forms of LDHA complexes with malonate (green) and NADH (orange) are shown. Opened active-site form of LDHA complex with machilin A (blue) is shown. The active-site loops (97–108) from each chains of LDHA- machilin A tetramer are superimposed. (**F**) Scheme about inhibitory of LDHA activity by machilin A.

**Figure 3 cancers-11-00963-f003:**
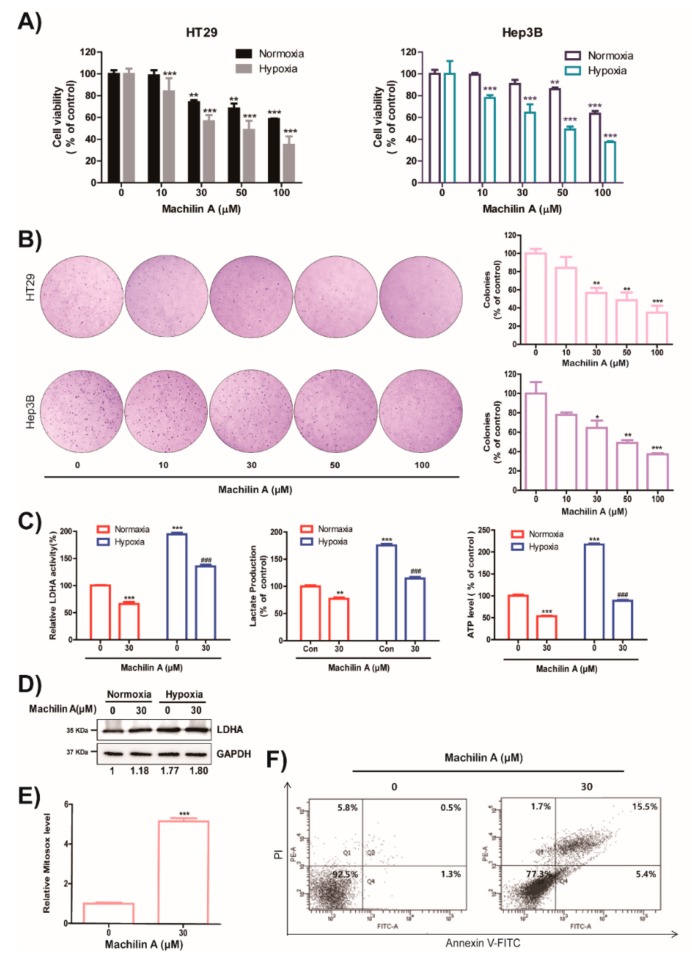
Effect of machilin A on cell growth, colony formation, Lactate dehydrogenase A (LDHA) activity, lactate production, intracellular adenosine triphosphate (ATP levels), and expression of LDHA in cancer cells. (**A**) Cancer cells were treated with the indicated concentrations of machilin A for 48 h under hypoxia. The viabilities of cells were evaluated using an MTT assay. Results of three independent replicates are shown as means ± SD. Statistic tests were performed using a one-way analysis of variance (ANOVA). *** *p* < 0.001 compared to the control. (**B**) To determine colony formation capacity of cancer cells, HT29 colon cancer and Hep3B liver cancer cells were treated with machilin A for three weeks, and colonies were counted. Results are presented as means ± SD. Statistic tests were performed using a one-way analysis of variance (ANOVA). * *p* < 0.05, ** *p* < 0.01, and *** *p* < 0.001, compared to the control. (**C**) HT29 colorectal cancer cells were treated with machilin A (30 μM) under normoxic and hypoxic conditions. Potential inhibition of LDH activity was assessed using HT29 cell lysates. Production of lactate was measured using a commercially available L-lactate assay kit. Results of three independent replicates are shown as means ± SD. Statistic tests were performed using a Student’s t-test. ** *p* < 0.01 and *** *p* < 0.001, compared to the control under normoxia (first column). ### *p* < 0.001 compared to control under hypoxia (third column). (**D**) Expressions of LDHA were analyzed using a western blot assay. GAPDH expression was used as an internal control. (**E**) The HT29 cells were treated with machilin A for 24 h. The production of intracellular reactive oxygen species (ROS) was detected by Fluorescence-activated cell sorting (FACS) analysis using mitochondrial ROS indicator, Mitosox. Results of three independent replicates are shown as means ± SD. Statistic tests were performed using a Student’s t-test. *** *p* < 0.001, compared to the control (first column). (**F**) The HT29 cells were treated with indicated concentrations of machilin A for 48 h. The cells were harvested, stained with Annexin V/PI, and analyzed by flow cytometry.

**Figure 4 cancers-11-00963-f004:**
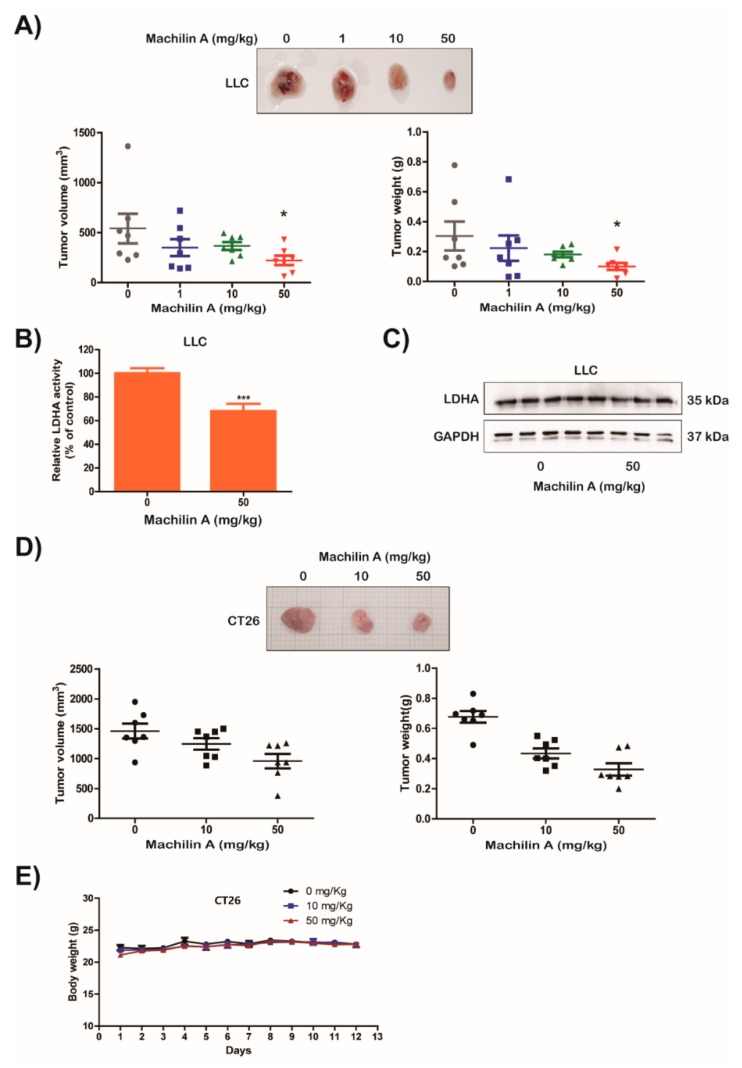
Suppression of tumor growth by machilin A in mice inoculated with lung and colon cancer cells. LLC and CT26 cells were inoculated subcutaneously on the back of C57BL/6 mice and BALB/c mice. After inoculation, the mice were intraperitoneally injected with the indicated dosages of machilin A on a daily basis. Mice were sacrificed 21 days after inoculation. (**A**,**D**) Photographs of each tumor treated with the indicated dose of machilin A are shown. Volume and weight of each tumor was measured. Data are presented as means ± SD. Statistic tests were performed using a Student’s t-test. * *p* < 0.05 compared to the control. Cancer tissues harvested from mice were lysed. (**B**) Lactate dehydrogenase A (LDHA) activity was analyzed. Data are presented as means ± SD. Statistic tests were performed using a Student’s t-test. *** *p* < 0.001 compared to the control. (**C**) The expression of LDHA was examined using a western blot analysis. GAPDH expression was used as an internal control. Data are shown after a densitometric analysis. (**E**) Body weight of each mouse was measured daily during injection of machilin A.

**Figure 5 cancers-11-00963-f005:**
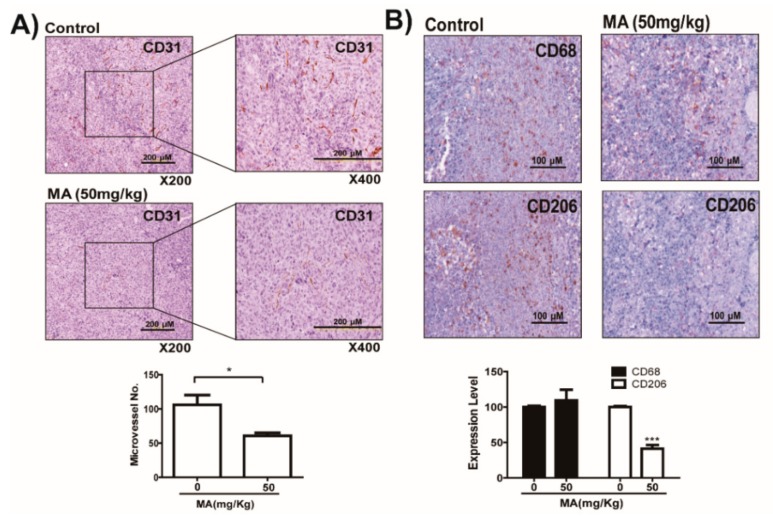
Inhibition of macrophage-mediated neovascularization by suppression of tumor-derived lactic acid in machilin A (MA)-treated cancer cells. (**A**) Tumor specimens were fixed in phosphate-buffered saline (PBS) with 3.7% formalin and then embedded in paraffin. Tumor tissues were immunostained using a CD31 antibody to detect newly formed vessels in tumor tissues of mice inoculated with LLC cells. Microvessels were manually counted. Data are presented as mean ± SD. Statistic tests were performed using a Student’s t-test. * *p* < 0.05 compared to the control. (**B**) Tumor tissues were immunostained using an antibody against CD206 M2 phenotype macrophage marker to detect M2 phenotype macrophages, and using an antibody against CD68 macrophage marker to detect infiltrated macrophages in tumor tissues in mice inoculated with LLC cells. Positive spots (brown color) were manually counted. Data are presented as mean ± SD. Statistic tests were performed using a Student’s t-test. *** *p* < 0.001 compared to the control. (**C**) Scheme about treatment of cancer conditioned media to macrophage. (**D**) HT29 cancer cells were treated with machilin A (30 µM) and incubated under normoxic or hypoxic conditions. After 48 h of incubation, the conditioned medium (1 mL) harvested from cancer cells was added to RAW264.7 macrophage cells in the presence or absence of lactate (5 mM). The respective expression of Arg-1, YM1, and VEGF mRNAs was determined using qRT-PCR. Data are presented as mean ± SD. Statistic tests were performed using a Student’s t-test. ** *p* < 0.01 and *** *p* < 0.001 compared to No. 1. ## *p* < 0.01 and ### *p* < 0.001 compared to No. 2. §§ *p* < 0.01 and §§§ *p* < 0.001 compared to No. 3. (**E**) Quantitative summary of tube formation was performed by manual quantification of closed tubes in four different areas, scale bar = 10 µm. Results of three independent replicates are shown as means ± SD. Statistic tests were performed using a Student’s t-test. ** *p* < 0.01 compared to No. 1. *** *p* < 0.001 compared to No. 2. ### *p* < 0.001 compared to No. 3.

**Table 1 cancers-11-00963-t001:** IC_50_ of machilin A (MA) on the cell viability and LDHA activity.

Cell Line	Cell Viability (μM)	LDHA Activity (μM)
WiDr	17.19 ± 5.75	61.87 ± 8.02
DLD-1	423.41 ± 17.55	129.46 ± 24.82
RKO	192.16 ± 15.19	73.67 ± 20.69
HT29	107.01 ± 4.66	48.42 ± 4.33
MCF-7	216.75 ± 12.88	61.18 ± 3.02
HCC-95	228.38 ± 13.93	112.34 ± 10.11
NCI-H1793	96.68 ± 5.32	63.99 ± 5.61
Huh7	198.44 ± 26.70	85.07 ± 20.90
HepG2	155.51 ± 23.24	100.78 ± 40.47
Hep3B	139.50 ± 8.55	48.30 ± 4.13

**LDHA:** Lactate dehydrogenase A.

**Table 2 cancers-11-00963-t002:** Inhibitory effect of M machilin A (MA) on the production of lactate.

Cell Line	MA 0 µM	MA 10 µM	MA 50 µM
	Lactate (mM)		
WiDr	3.37 ± 0.06	3.22 ± 0.05	2.70 ± 0.06
DLD-1	4.74 ± 0.05	3.68 ± 0.09	2.94 ± 0.17
R.K.O	2.94 ± 0.07	2.76 ± 0.06	2.48 ± 0.06
HT29	3.68 ± 0.18	3.42 ± 0.37	2.73 ± 0.04
MCF7	1.13 ± 0.14	1.07 ± 0.15	0.77 ± 0.19
HCC-95	2.68 ± 0.09	2.25 ± 0.16	1.76 ± 0.10
NCI-H1793	2.72 ± 0.48	2.36 ± 0.26	2.04 ± 0.38
Huh7	1.58 ± 0.30	1.14 ± 0.32	0.67 ± 0.15
HepG2	0.98 ± 0.20	0.51 ± 0.06	0.47 ± 0.07
Hep3B	1.20 ± 0.02	1.12 ± 0.04	0.73 ± 0.01

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
