# Peer review of "Machilin A Inhibits Tumor Growth and Macrophage M2 Polarization Through the Reduction of Lactic Acid"

_cancers, 2019, doi:10.3390/cancers11070963_

Round 1

Reviewer 1 Report

The authors performed a screen of 480 compounds from a natural library for LDHA, a key enzyme and driver of the Warburg effect in cancer. The compound machilin A (MA) exhibited the strongest inhibition of LDHA with micromolar IC50. They go on to solve the crystal structure of MA bound to LDHA compared to malonate/NADH showing a switch between open and closed forms of the enzyme. MA showed a significant inhibition of cell growth, reduction in cellular LDHA activity, reduced lactate production and decreased ATP. In addition in vivo MA reduced tumour volume and microvessel density via a mechanism involving decreased macrophage recruitment. Overall this is a well performed study containing a wealth of data showing the structural elucidation of it's binding and the mechanism of action both in vitro and in vivo. I have some comments that might be considered or expanded upon.

1) Please comment on the potential use of MA for therapy or combination therapy in humans?

2) Can the authors comment on possible off target activity of MA against other enzymes. For example it is reported (Phytomedicine 22, 2015, 615-620) to inhibit cytochrome P450 with IC50 of the order 3 micromol. Thus there is potential for biological effects that are not specific to LDHA. 

3) Related to point 2 would the same biological effects of MA be recapitulated by knockdown of LDHA by siRNA or by another small molecule inhibitor thus confirming the effects seen with MA?  

Reviewer 2 Report

The article by  Ghung et al presents interesting data on machilin A, a novel inhibitor of Lactate dehydrogenase A (LDHA).  LDHA is an important enzyme in fermentative glycolysis, generating most energy for cancer cells that rely on anaerobic respiration even under normal oxygen concentrations. This renders LDHA a promising molecular target for the treatment of various cancers. Several efforts have been made recently to develop LDHA inhibitors.

This manuscript contains experimental data with interesting implications, but I believe need some clarifications in order to understand the relevance of the data presented.

In order to translate these results to therapeutic level, the authors should show whether MA has an effect on the viability of non-tumoral cells.

Please report the experimental condition for hypoxic  conditions (%O2 and time of exposure in  hypoxia).

In Table 1 please change cell viability (uM) in IC50 (uM).  Are the results  shown in Table 1 obtained in hypoxic or normoxic conditions?.

Was this inhibitor able to induce cell death even in in vivo tumors?

Did the effect of this inhibition continue even when the drug is removed from the cells?

What is the role of MA on the expression and activity of the other LDHB isoform in cancer cells?.

The determination of lactate content was reported for the extracellular lactate level . Did they measure the intracellular lactate content?

Please report on discussion a recent paper by Kim et al  A Novel Lactate Dehydrogenase Inhibitor, 1-(Phenylseleno)-4-(Trifluoromethyl) Benzene, Suppresses Tumor Growth through Apoptotic Cell Death.. Sci Rep. 2019 Mar 8;9(1):3969. doi: 10.1038/s41598-019-40617-3.

I believe that it is an interesting paper, suitable for publication in Cancers, after major revisions.

Round 2

Reviewer 2 Report

The authors clarified several points and improved the text.

I think it's an interesting article for readers of the Journal.